# Converting Simple Temporal Networks with Uncertainty into Minimal Equivalent Dispatchable Networks

**Primary Keywords:** *(6) Temporal Planning;*

## Abstract

A *Simple Temporal Network with Uncertainty* (STNU) is a structure for representing and reasoning about time constraints on actions with uncertain durations. An STNU is *dynamically controllable* (DC) if there exists a dynamic strategy for executing the network that guarantees that all of its constraints will be satisfied no matter how the uncertain durations turn out—within their specified bounds. However, such strategies typically require exponential space. Therefore, it is essential to convert a DC STNU into a so-called *dispatchable* form for practical applications. For a dispatchable STNU, the relevant portions of a real-time execution strategy can be incrementally constructed during execution, requiring only $O(n^2)$ space, while also providing maximum flexibility but requiring only minimal computation during execution. Existing algorithms can generate equivalent dispatchable STNUs, but with no guarantee about the number of edges in the output STNU. Since that number directly impacts the computations during execution, this paper presents a novel algorithm for converting any dispatchable STNU into an equivalent dispatchable network having a minimal number of edges. The complexity of the algorithm is $O(kn^3)$, where $k$ is the number of actions with uncertain durations, and $n$ is the number of timepoints. The paper also provides an empirical evaluation of the order-of-magnitude reduction of edges obtained by the new algorithm.

## 1   Introduction and Related Work

Temporal constraint networks facilitate representing and reasoning about temporal constraints on activities. Among the many kinds of temporal constraint networks in the literature, *Simple Temporal Networks with Uncertainty* (STNUs) are one of the most important because they allow the explicit representation of actions with uncertain durations (Morris, Muscettola, and Vidal 2001). In an STNU, an action with uncertain duration is represented by a *contingent link* that specifies bounds on the duration's uncertainty. A contingent link also represents that a scheduler cannot *decide* the action's duration, but instead can only *observe* it at runtime. Therefore, for STNUs, it is important to know whether a strategy exists for executing the network that guarantees that all of its constraints will be satisfied no matter how the uncertain durations play out. If such a strategy exists, the STNU is said to be *dynamically controllable* (DC). The literature includes several polynomial-time algorithms for checking the DC property that differ in their approaches to finding possible inconsistencies, characterized by different kinds of negative cycles (Stedl and Williams 2005; Morris 2006, 2014; Nilsson, Kvarnstrom, and Doherty 2014; Cairo, Hunsberger, and Rizzi 2018). But the algorithms are not constructive (i.e., they do not output actual execution strategies in positive instances).

For DC STNUs, an execution strategy can be specified by pre-computing every possible incremental schedule for every possible combination of uncertain action durations. However, such a strategy requires exponential space and, therefore, is exceedingly impractical. So, for practical applications, Morris (2014, 2016) proposed converting DC STNUs into an equivalent *dispatchable* form. For a dispatchable STNU, a dynamic execution strategy can be generated incrementally, during execution, using only $O(n^2)$ space, where $n$ is the number of timepoints (citation omitted/blind review). Moreover, such a strategy provides maximum flexibility for the action scheduler, but requires minimal real-time computation.

Most DC-checking algorithms do not guarantee a dispatchable output, but Morris (2014) indicated that his DC-checking algorithm could be modified to do so. We call that version of his algorithm *Morris14*. More recently, Hunsberger and Posenato (2023) presented an algorithm for creating an equivalent dispatchable STNU, called $FD_{\text{STNU}}$. Although these are the only known algorithms that guarantee an equivalent dispatchable output, they do not provide any guarantees about the *number of edges* in it. Since the number of edges directly impacts computations during execution, this paper presents a novel algorithm for converting any dispatchable STNU into an equivalent dispatchable network with a *minimal* number of edges. The complexity of the algorithm is $O(kn^3)$, where $k$ is the number of actions with uncertain durations, and $n$ is the number of nodes in the network. The paper provides an empirical evaluation of the *order-of-magnitude* reduction in the number of edges obtained by the new algorithm.

## 2   Background

This section summarizes definitions and results for the dispatchability of Simple Temporal Networks (STNs) and STNs with Uncertainty (STNUs). Our focus is on the dispatchability of STNUs, but the work on STNs is also important.

**Definition 1** ((Dechter, Meiri, and Pearl 1991)). A *Simple Temporal Network* (STN) is a pair $\mathcal{S} = (\mathcal{T}, \mathcal{C})$, where $\mathcal{T}$ is a set of real-valued variables called *timepoints* (TPs) and

$\mathcal{C}$ is a set of constraints of the form $(Y - X \leq \delta)$, for some $X, Y \in \mathcal{T}$ and some $\delta \in \mathbb{R}$. Each STN $\mathcal{S} = (\mathcal{T}, \mathcal{C})$ has a corresponding graph $\mathcal{G} = (\mathcal{T}, \mathcal{E})$, where the TPs in $\mathcal{T}$ serve as the graph's nodes, and the constraints in $\mathcal{C}$ correspond to labeled, directed edges in $\mathcal{E}$. In particular: $\mathcal{E} = \{X \xrightarrow{\delta} Y \mid (Y - X \leq \delta) \in \mathcal{C}\}$. For convenience, $X \xrightarrow{\delta} Y$ may be notated as $(X, \delta, Y)$.

An STN is *consistent* if it has a solution as a constraint satisfaction problem. Although solutions for consistent STNs can be computed in advance of execution (e.g., by the *Bellman-Ford* algorithm (Cormen et al. 2022)), it is often desirable to preserve as much flexibility as possible during execution (e.g., to enable reacting to unanticipated events). Toward that end, Tsamardinos, Muscettola, and Morris (1998) first specified a real-time execution algorithm for STNs, called the *Time Dispatching* (TD) algorithm, and then defined an STN to be *dispatchable* if every run of the TD algorithm was guaranteed to generate a solution. The TD algorithm provides maximum flexibility during execution by maintaining *time windows* for each timepoint. It requires minimal computation during execution by propagating the effects of each real-time execution, $X = v$, only *locally* (i.e., to $X$'s *neighbors*; that is, timepoints connected to $X$ by a single edge). Morris (2016) subsequently provided a graphical characterization of STN dispatchability in terms of *vee-paths*.

**Definition 2** (Vee-path (Morris 2016))**.** A *vee-path* is a path consisting of zero or more negative edges followed by zero or more non-negative edges. If $\mathcal{P}$ is a vee-path from $X$ to $Y$ that is also a shortest path from $X$ to $Y$, then $\mathcal{P}$ is called a *shortest vee-path* (SVP). If, in addition, $\mathcal{P}$ contains no proper cycles, then $\mathcal{P}$ is called a *simple shortest vee-path* (SSVP).

**Theorem 1.** *(Morris 2016) An STN is dispatchable if and only if whenever there is a path from any $X$ to any $Y$, then there is a shortest vee-path from $X$ to $Y$ (i.e., an SVP).*

**Definition 3** ((Morris, Muscettola, and Vidal 2001))**.** A *Simple Temporal Network with Uncertainty* (STNU) augments an STN to accommodate actions with uncertain durations. Formally, an STNU is a triple, $\mathcal{S} = (\mathcal{T}, \mathcal{C}, \mathcal{L})$, where $(\mathcal{T}, \mathcal{C})$ is an STN, and $\mathcal{L}$ is a set of *contingent links* (CLs), each of the form $(A, x, y, C)$, where $A, C \in \mathcal{T}$ and $0 < x < y < \infty$.

Intuitively, once the *activation timepoint $A$* is *executed* (i.e., assigned a value during execution), the *contingent timepoint $C$* is guaranteed to be executed such that the duration $C - A \in [x, y]$, but the particular execution time for $C$ is not controlled by the agent in charge of executing the network; instead, it is only *observed* in real time.

$\Rightarrow$ For convenience, and without loss of generality, we assume that no contingent timepoint $C$ can serve as the activation timepoint for another contingent link.

Each STNU $\mathcal{S} = (\mathcal{T}, \mathcal{C}, \mathcal{L})$ has a corresponding graph, $\mathcal{G} = (\mathcal{T}, \mathcal{E}_o \cup \mathcal{E}_{lc} \cup \mathcal{E}_{uc})$, where $(\mathcal{T}, \mathcal{E}_o)$ is the graph for the STN $(\mathcal{T}, \mathcal{C})$, and $\mathcal{E}_{lc}$ and $\mathcal{E}_{uc}$ contain labeled, directed edges derived from the contingent links in $\mathcal{L}$. In particular: $\mathcal{E}_{lc} = \{A \xrightarrow{c:x} C \mid (A, x, y, C) \in \mathcal{L}\}$, and $\mathcal{E}_{uc} = \{C \xrightarrow{C:-y} A \mid (A, x, y, C) \in \mathcal{L}\}$. The so-called *lower-case* (LC) edge $A \xrightarrow{c:x} C$ represents the uncontrollable possibility that the duration $C - A$ might take on its minimum value $x$, while

the so-called *upper-case* (UC) edge $C \xrightarrow{C:-y} A$ represents the uncontrollable possibility that $C - A$ might take on its maximum value $y$. Such edges may be respectively notated as $(A, c:x, C)$ and $(C, C:-y, A)$, while constraints in $\mathcal{C}$ and edges in $\mathcal{E}_o$ may be called *ordinary* constraints and edges, respectively, to distinguish them from the LC and UC edges.

**Definition 4.** An STNU $\mathcal{S} = (\mathcal{T}, \mathcal{C}, \mathcal{L})$ is *dynamically controllable* (DC) if there exists a *dynamic strategy* for executing its timepoints such that all constraints in $\mathcal{C}$ are guaranteed to be satisfied no matter how the durations of the CLs in $\mathcal{L}$ turn out (Morris, Muscettola, and Vidal 2001). The strategy is *dynamic* in that its execution decisions cannot depend on advance knowledge of future contingent executions.

Several polynomial-time *DC-checking* algorithms have been presented in the literature (Morris 2006, 2014; Cairo, Hunsberger, and Rizzi 2018; Hunsberger and Posenato 2022). However, in positive instances, such algorithms only confirm the *existence* of a dynamic execution strategy; they do not output one. Since such strategies typically require exponential space, Morris (2016) extended the concept of *dispatchability* from STNs to *extended* STNUs (ESTNUs), as follows.

First, we must backtrack. Some of the DC-checking algorithms for STNUs generate a new kind of *conditional* constraint called a *wait* (Morris 2006). A typical wait can be glossed as: *"If the contingent timepoint $C$ has not yet executed, then $W$ must wait until at least $w$ after the activation timepoint $A$."* Its graphical representation is the *generated* UC edge, $W \xrightarrow{C:-w} A$, where $A$ and $C$ are the activation and contingent timepoints for some CL $(A, x, y, C)$. Intuitively, since the execution of $C$ cannot be directly controlled and might occur as late as $y$ after $A$, $W$ must wait until $w$ after $A$; but if $C$ executes earlier than $w$ after $A$, then the wait is automatically satisfied and $W$ may be executed immediately.

Generating wait edges is not required to determine the DC property (e.g., as seen in the algorithms of Morris (2014) and Cairo, Hunsberger, and Rizzi (2018)), but wait edges turn out to be necessary for enforcing the dispatchability of STNUs. Anticipating this, Morris (2016) defined an *extended* STNU (ESTNU) to be an STNU augmented to include zero or more waits (equivalently, an STNU graph together with a set $\mathcal{E}_{ucg}$ of *generated* UC edges). He then defined the dispatchability of ESTNUs in terms of their STN *projections*, as follows.

**Definition 5** (Situation)**.** Let $\mathcal{S}$ be an STNU with $k$ contingent links whose duration ranges are $[x_1, y_1], \ldots, [x_k, y_k]$. A *situation* (for $\mathcal{S}$) is a $k$-tuple $\omega = (\omega_1, \omega_2, \ldots, \omega_k)$ where $\omega_i \in [x_i, y_i]$ for each $i \in \{1, 2, \ldots, k\}$. The space of all situations is notated as $\Omega = [x_1, y_1] \times \ldots \times [x_k, y_k]$.

In other words, a situation $\omega \in \Omega$ specifies a fixed duration for each contingent link. For convenience, if $C$ is the contingent timepoint for a link $(A, x, y, C)$, then the duration $C - A$ in the situation $\omega$ may be notated as $\omega_c$.

**Definition 6** (Projection)**.** For any ESTNU $\mathcal{G} = (\mathcal{T}, \mathcal{E}_o \cup \mathcal{E}_{lc} \cup \mathcal{E}_{uc} \cup \mathcal{E}_{ucg})$, and any situation $\omega$, the *projection* of $\mathcal{G}$ onto $\omega$ is the **STN** $\mathcal{G}_\omega = (\mathcal{T}, \mathcal{E}_o \cup \mathcal{E}_{lc}^\omega \cup \mathcal{E}_{uc}^\omega \cup \mathcal{E}_{ucg}^\omega)$, where:

- $\mathcal{E}_{lc}^\omega = \{(A_i, \omega_i, C_i) \mid \exists (A_i, c_i{:}x_i, C_i) \in \mathcal{E}_{lc}\}$
- $\mathcal{E}_{uc}^\omega = \{(C_i, -\omega_i, A_i) \mid \exists (C_i, C_i{:}-y_i, A_i) \in \mathcal{E}_{uc}\}$

- $\mathcal{E}_{\text{ucg}}^{\omega} = \{(W, \delta, A_i) \mid \exists (W, C_i{:}{-}v, A_i) \in \mathcal{E}_{\text{ucg}}$
  and $\delta = \max\{-\omega_i, -v\}\}$

**Definition 7** (Dispatchable ESTNU). An ESTNU is dispatchable iff all of its STN projections are dispatchable (as STNs).

Morris suggested that a dispatchable ESTNU could be executed using the TD algorithm for STNs, while pretending that the execution times for the contingent TPs were chosen by external reality. Others (citation omitted for blind review) recently confirmed this by formally defining a real-time execution algorithm for ESTNUs and proving that it is guaranteed to successfully execute any dispatchable ESTNU, preserving maximum flexibility while requiring minimal computation.

Most DC-checking algorithms do not output a dispatchable ESTNU. However, as Morris (2014) noted, his $O(n^3)$-time DC-checking algorithm can easily be modified to generate waits and, insodoing, guarantee that it will output an equivalent dispatchable ESTNU. More recently, Hunsberger and Posenato (2023) presented a faster algorithm for generating equivalent dispatchable ESTNUs. However, neither algorithm makes any claim about the number of edges in the dispatchable output. Since that number impacts the performance of the real-time execution algorithm, producing an equivalent dispatchable ESTNU with a minimal number of edges is of practical importance. This paper presents the first algorithm for solving this problem. It runs in $O(kn^3)$ worst-case time.

## 3  The *minDisp*ESTNU Algorithm

This section introduces our new *minDisp*ESTNU algorithm. When given a dispatchable ESTNU as input, it outputs an equivalent dispatchable ESTNU with a minimal number of edges.[1] Since the input is dispatchable, the algorithm need only determine which edges can be removed while preserving dispatchability (i.e., while ensuring that every pair of timepoints connected by a path are connected by an SVP in every projection). However, a key observation is that for any timepoints $V$ and $Y$, the length of the shortest vee-path from $V$ to $Y$ may be different across different STN projections.

**Definition 8** (Path-length notation). The length of a path $\mathcal{P}$ in a projection $\mathcal{G}_\omega$ is notated as $|\mathcal{P}|_\omega$. For timepoints $X$ and $Y$, $d_\omega(X, Y)$ denotes the length of the shortest path from $X$ to $Y$ in $\mathcal{G}_\omega$; and $\text{maxd}(X, Y) = \max_\omega\{d_\omega(X, Y)\}$ denotes the maximum such length *across all projections*. Finally, $d_o(X, Y)$ denotes the length of the shortest path from $X$ to $Y$ that has only *ordinary* edges from the ESTNU graph.

For example, suppose $e = (V, \sigma, Y)$ is an ordinary edge in a dispatchable ESTNU $\mathcal{G}$. Suppose further that in every STN projection $\mathcal{G}_\omega$, there is an SVP $\mathcal{P}$ from $V$ to $Y$ that does *not* use $e$, and such that $|\mathcal{P}|_\omega \leq \sigma$. Then in every projection, $e$ is not needed to ensure dispatchability. Therefore, the ESTNU obtained by removing $e$ must still be dispatchable.

If $\text{maxd}(V, Y)$ is determined solely by ordinary edges, then the dispatchability algorithm for STNs (i.e., for ordinary edges) can be used to remove all ordinary edges dominated by ordinary vee-paths (Tsamardinos, Muscettola, and Morris

---

[1]We intentionally blur the distinction between an ESTNU and its graph, and between edges and constraints.

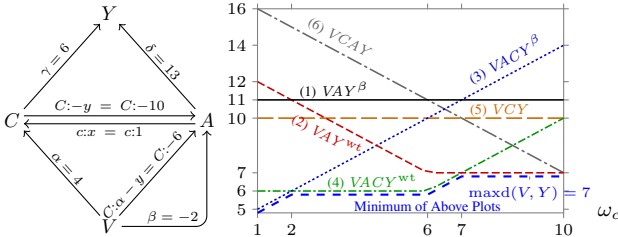

Figure 1: Special structure that determines $\text{maxd}(V, Y)$

1998). However, the value of $\text{maxd}(V, Y)$ can also be determined by a variety of paths involving *labeled edges* (e.g., LC, UC or wait edges) *whose lengths differ across projections.*

The left of Figure 1 illustrates the kind of *special structure* that can determine the value of $\text{maxd}(V, Y)$ based on different paths comprising various combinations of ordinary and labeled edges. Since this example has only one contingent link, $(A, x, y, C) = (A, 1, 10, C)$, the range of projections is given by $\omega_c \in [x, y] = [1, 10]$. Particularly relevant is that in different projections the length of the shortest vee-path from $V$ to $Y$ is determined by different paths. For example, the blue dotted line in the plot on the right of the figure shows the length of the path $VACY^\beta$, where the superscript $\beta$ signals that the first edge in this path is the ordinary edge $(V, \beta, A) = (V, -2, A)$ from $V$ to $A$, not the *wait* edge. This path, which also includes the LC edge $(A, c{:}x, C) = (A, c{:}1, C)$, happens to be shortest in projections where $\omega_c \in [1, 2]$. In contrast, the green dot-dashed line shows the length of the path $VACY^{wt}$ that instead uses the *wait* edge, $(V, C{:}\alpha - y, A) = (V, C{:}-6, A)$. This path is shortest in projections where $\omega_c \in [2, 7]$. Finally, the dashed red line shows the length of the path $VAY^{wt}$, which also uses the wait edge. This path is shortest for $\omega_c \in [7, 10]$. The black and dashed-orange lines indicate the lengths of the ordinary paths $VAY$ and $VCY$, which are not shortest in any projection. The dot-dashed gray line indicates the length of $VCAY$, which uses the UC edge $(C, C{:}-10, A)$ and is only shortest in the projection where $\omega_c = y = 10$. The thick, blue dashed line plots the lengths of the *shortest* vee-paths from $V$ to $Y$ across all projections. Its maximum value is 7; hence, $\text{maxd}(V, Y) = 7$. In other words, despite there being no ordinary path from $V$ to $Y$ of length 7, the edges in this special structure nonetheless entail the constraint $Y - V \leq 7$.

The lengths of the relevant paths in Figure 1 are specified algebraically in Table 1. Since each plot is continuous, piecewise-linear and monotonic (whether non-increasing or non-decreasing), the minimum plot, represented by the thick blue dashed line in Figure 1, can be determined by finding each value $\omega_c^\dagger \in [x, y] = [1, 10]$ at which any of the above plots might intersect, and then taking the minimum value of the plots at each such $\omega_c^\dagger$. The *maximum* of those *minimum* values, across all of the intersection points, will be the maximum SVP length across all projections (i.e., $\text{maxd}(V, Y)$).

### The Problem Turns Out to be Simpler!

Although six different paths from $V$ to $Y$ in the ESTNU graph from Figure 1 were explored above, it turns out that in

(1) $|VAY^{\beta}|_{\omega_c} = \beta + \delta = 11$

(2) $|VAY^{wt}|_{\omega_c} = \max\{-\omega_c, \alpha - y\} + \delta$
$= \max\{\delta - \omega_c, \alpha + \delta - y\} = \max\{13 - \omega_c, 7\} \in [7, 12]$

(3) $|VACY^{\beta}|_{\omega_c} = \beta + \omega_c + \gamma = 4 + \omega_c \in [5, 14]$

(4) $|VACY^{wt}|_{\omega_c} = \max\{-\omega_c, \alpha - y\} + \omega_c + \gamma$
$= \max\{\gamma, \alpha + \gamma - y + \omega_c\} = \max\{6, \omega_c\} \in [6, 10]$

(5) $|VCY|_{\omega_c} = \alpha + \gamma = 10$

(6) $|VCAY|_{\omega_c} = \alpha - \omega_c + \delta = 17 - \omega_c \in [7, 16]$

Table 1: The lengths of the paths in Figure 1

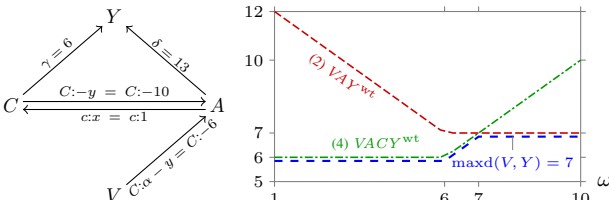

Figure 2: A simpler structure determines $\mathrm{maxd}(V, Y)$

general it suffices to consider only *two* of the paths: namely, (2) $VAY^{wt}$ and (4) $VACY^{wt}$, as shown in Figure 2. As will be seen, the desired max-of-the-mins value must occur at the intersection of these two paths. But if the intersection point is outside of the interval $(x, y)$, then one of the paths, (2) and (4), will dominate the other throughout the entire interval $[x, y]$ which, as will be shown, implies that a pre-existing ordinary path is shortest from $V$ to $Y$. On the other hand, if the intersection lies within the interval $(x, y)$, the desired $\mathrm{maxd}(V, Y)$ value will necessarily occur at that intersection point—because path (2) is non-increasing, while path (4) is non-decreasing. Hence, it suffices to find where the paths (2) and (4) intersect, which a simple computation reveals is where $\omega = \delta - \gamma$. For the numbers in Figure 2, the two paths intersect where $\omega = \delta - \gamma = 13 - 6 = 7 \in (1, 10)$. Their lengths in the projection, $\omega = 7$, gives the value of $\mathrm{maxd}(V, Y)$, which also happens to be 7, as seen earlier.

Although the path, (3) $VACY^{\beta}$, in Figure 1 has the minimum length for $\omega \in [1, 2]$, that has no effect on the value of $\mathrm{maxd}(V, Y)$. Our new algorithm explores simpler structures like those in Figure 2 to find the strongest constraints entailed by various combinations of labeled and ordinary edges.

## Nesting of Special Structures

The ESTNU in Figure 3 contains two special structures, one nested inside the other. The inner structure involves the timepoints $A_2, A, C$ and $Y$; the outer structure involves $V_2, A_2, C_2$ and $Y$. The values of $\delta, \gamma$ and $\hat{\omega}$ for the outer structure, using the ordinary edge $(A_2, 9, Y)$ as the path from $A_2$ to $Y$, and the ordinary edge $(C_2, 2, Y)$ as the path from $C_2$ to $Y$, are: $\delta = 9, \gamma = 2$, and $\hat{\omega} = 9 - 2 = 7$. This implies an entailed ordinary edge $(V_2, 3, Y)$, shown as dashed and red in the figure. The inner structure, as seen in Figure 2, entails the ordinary edge $(A_2, 7, Y)$, also shown as red and dashed. The key point is that with the entailed edge $(A_2, 7, Y)$, a

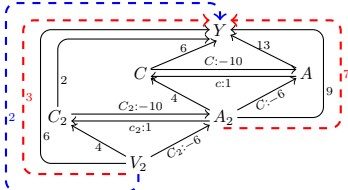

Figure 3: Example of nested of special structures

new analysis of the outer structure reveals stronger values ($\delta = 7, \gamma = 2$ and $\hat{\omega} = 7 - 2 = 5$) that entail a stronger edge $(V_2, 2, Y)$, shown as blue and dashed in the figure.

Since special structures can be nested, exploring each one only once will typically not be sufficient. However, as will be shown later on, it suffices to consider nestings that use the labeled edges from each contingent link at most once. Therefore, $k$ rounds of processing, where each round involves processing every special structure, will suffice.

## Algorithm Pseudocode

Algorithm 1 shows pseudocode for our new algorithm. It takes a dispatchable ESTNU as its only input and generates as its output an equivalent dispatchable ESTNU having a minimal number of edges. The algorithm begins (at Lines 3-8) by inserting any missing *weak-ord edges* needed to make the input ESTNU *weak-ord complete* (defined below).

**Definition 9** (Weak-ord Edges). Let $(A, c{:}x, C)$, $(C, C{:}{-}y, A)$ and $(V, C{:}{-}v, A)$ be the LC, UC and any wait edges associated with a contingent link $(A, x, y, C)$. The corresponding *weak-ord edges* are $(A, y, C), (C, -x, A)$ and $(V, -x, A)$. For any labeled edge $e$, the corresponding weak-ord edge is denoted by $e_o$. An ESTNU $\mathcal{G}$ is called *weak-ord complete* if for *each* labeled edge $e$ in $\mathcal{G}$, $\mathcal{G}$ contains an ordinary edge $\hat{e}$ such that $|\hat{e}| \leq |e_o|$.

Note that for any edge $e$ and situation $\omega$, $|e|_\omega \leq |e_o|$. For the LC and original UC edge, the result is trivial. For waits: $|(V, C{:}{-}v, A)|_\omega = \max\{-v, -\omega_c\} \leq \max\{-x, -x\} = -x = |(W, -x, A)|$. (We assume $-v < -x$, since otherwise the wait is equivalent to the ordinary edge $(V, -x, A)$.)

Since every inserted weak-ord edge is dominated by the projection of its labeled counterpart in $\mathcal{G}$, such edges are not needed for dispatchability. However, as will be seen, making them explicit in $\mathcal{G}$ enables the algorithm to compute $\mathrm{maxd}(V, Y)$ values derived from the kinds of special structures seen in Figure 2. To ensure that they are removed before the end of the algorithm, each inserted weak-ord edge is "marked" for later removal.

The next part of the algorithm comprises the multiply nested loops at Lines 9-23. Their purpose is to discover and insert ordinary edges that are entailed by the kinds of special structures seen in Figure 2. Since these structures can be nested to a maximum depth of $k$, the outermost loop does $k$ iterations. Each iteration begins (at Line 10) by using Johnson's algorithm to compute the *distance matrix* $d$ for the STN consisting of the ordinary edges from the input ESTNU—which includes the weak-ord edges inserted at Lines 3-8, as

**Algorithm 1:** $minDisp_{\text{ESTNU}}$

**Input:** $\mathcal{G} = (\mathcal{T}_x \cup \mathcal{T}_c, \ \mathcal{E}_{\text{o}} \cup \mathcal{E}_{\text{lc}} \cup \mathcal{E}_{\text{uc}} \cup \mathcal{E}_{\text{ucg}})$,
    a dispatchable ESTNU

**Output:** $\mathcal{G}$, *modified* to be an equivalent dispatchable ESTNU
    having a minimal number of edges

1   $\mathcal{L} :=$ the contingent links associated with $\mathcal{G}$
2   $\mathcal{T} := \mathcal{T}_x \cup \mathcal{T}_c; \ marked := \emptyset$
    // Insert weak-ord edges entailed by LC and UC edges; mark them
3   **foreach** $(A, x, y, C) \in \mathcal{L}$ **do**
4      $\mathcal{E}_{\text{o}} := \mathcal{E}_{\text{o}} \cup \{(A, y, C), (C, -x, A)\}$
5      $marked := marked \cup \{(A, y, C), (C, -x, A)\}$

    // Insert weak-ord edges entailed by wait edges; mark them
6   **foreach** $(V, C{:}{-}v, A) \in \mathcal{E}_{\text{ucg}}$ **do**
7      $\mathcal{E}_{\text{o}} := \mathcal{E}_{\text{o}} \cup \{(V, -x, A)\}$ // $x$ is lower bound for $C - A$
8      $marked := marked \cup \{(V, -x, A)\}$

9   **for** $i := 1$ **to** $k$   **do** // $k$ = max depth of nested special structures
10     $d := \text{Johnson}(\mathcal{T}, \mathcal{E}_{\text{o}})$ // Distance matrix for ord edges
     // Explore special structures involving any $V, A, C$ and $Y$
11     $edgeAdded := \bot$
12     **foreach** $(A, x, y, C) \in \mathcal{L}$ **do**
13       **foreach** $V, Y \in \mathcal{T} \backslash \{A, C\} \mid V \not\equiv Y$ **and**
        $\exists (V, C{:}{-}q, A) \in \mathcal{E}_{\text{ucg}}$ **do**
14        $\gamma := d(C, Y), \delta := d(A, Y)$
15        **if** $\gamma < \infty$ **and** $\delta < \infty$ **then**
16         $\hat{\omega} := \delta - \gamma$
17         **if** $x < \hat{\omega} < y$ **then**
18          $\theta := \max\{-\hat{\omega}, -q\} + \delta$
19          **if** $\theta \leq d(V, Y)$ **then** // max min ord edge
20           $\mathcal{E}_{\text{o}} := \mathcal{E}_{\text{o}} \cup \{(V, \theta, Y)\}$
21           $marked := marked \cup \{(V, \theta, Y)\}$
22           $edgeAdded := \top$

23     **if** $edgeAdded == \bot$ **then** exit from the **for** loop

24   $\mathcal{E}_{\text{o}} := disp_{stn}(\mathcal{T}, \mathcal{E}_{\text{o}})$ // STN dispatchability on ordinary edges
25   **if** $edgeAdded$ **then** $d := \text{Johnson}(\mathcal{T}, \mathcal{E}_{\text{o}})$   // Update distances
26   $\mathcal{E}_{\text{o}} := \mathcal{E}_{\text{o}} \backslash marked$ // Remove marked edges from $\mathcal{E}_{\text{o}}$
27   $marked_{uc} := \emptyset$
28   **foreach** $C \in \mathcal{T}_c$ **do**
     // Remove negative ord. edges emanating from contingent TPs
29     **foreach** $(C, v, X) \in \mathcal{E}_{\text{o}}$ **do**
30       **if** $v < 0$ **then** $\mathcal{E}_{\text{o}} := \mathcal{E}_{\text{o}} \backslash \{(C, v, X)\}$
     // Remove non-negative ord edges terminating at contingent TPs
31     **foreach** $(X, v, C) \in \mathcal{E}_{\text{o}}$ **do**
32       **if** $v \geq 0$ **then** $\mathcal{E}_{\text{o}} := \mathcal{E}_{\text{o}} \backslash \{(X, v, C)\}$
     // Mark wait edges emanating from contingent TPs
33     **foreach** $(C, \hat{C}{:}v, \hat{A}) \in \mathcal{E}_{\text{ucg}} \mid C \in \mathcal{T}_c$ **do**
34       $marked_{uc} := marked_{uc} \cup \{(C, \hat{C}{:}v, \hat{A})\}$

35   **foreach** $(V, C{:}{-}v, A) \in \mathcal{E}_{\text{ucg}}$ **do** // Mark dominated waits from $\mathcal{E}_{\text{ucg}}$
36     **if** $d(V, A) \leq -v$ **or** $d(V, C) \leq 0$ **then**
37       $marked_{uc} := marked_{uc} \cup \{(V, C{:}{-}v, A)\}$
38     **else**
39       **foreach** $U \in \mathcal{T} \mid \exists (U, C{:}{-}u, A) \in \mathcal{E}_{\text{ucg}}$ **do**
40        **if** $d(V, U) \leq 0$ **and**
        $d(V, U) - u \leq \max\{-y, -v\}$ **then**
41         $marked_{uc} := marked_{uc} \cup \{(V, C{:}{-}v, A)\}$

42   $\mathcal{E}_{\text{ucg}} := \mathcal{E}_{\text{ucg}} \backslash marked_{uc}$
43   **return** $\mathcal{G} = (\mathcal{T}, \mathcal{E}_{\text{o}} \cup \mathcal{E}_{\text{lc}} \cup \mathcal{E}_{\text{uc}} \cup \mathcal{E}_{\text{ucg}})$

---

well as any ordinary edges inserted during previous iterations. The inner loops (Lines 12-22) then explore all possible simpler structures involving timepoints $V, A, C$ and $Y$, as illustrated in Figure 2. If the values of $\gamma$ and $\delta$ are finite, then $\hat{\omega} = \delta - \gamma$ is the value of $\omega_c$ at which the paths labeled (2) and (4) in Figure 2 intersect. If $\hat{\omega} \in (x, y)$, then the desired *max-of-the-mins* value (i.e., a possible update for $\text{maxd}(V, Y)$) (called $\theta$ in the pseudocode) is computed (Line 18). If $\theta \leq d(V, Y)$, then the ordinary edge $(V, \theta, Y)$ is inserted—and marked—to make explicit the constraint entailed by the special structure. It is marked because it is entailed by other edges and, hence, must be removed before the algorithm completes. In the meantime, it contributes to the distance matrix computation at the start of the next iteration.

After all of the ordinary edges entailed by combinations of labeled and ordinary edges have been inserted into the graph, the final part of the algorithm (Lines 24-42) determines which edges to *remove* from the ESTNU to achieve the minimal equivalent dispatchable ESTNU. At Line 24, the *STN* dispatchability algorithm is run on the subgraph of *ordinary* edges (i.e., the edges in $\mathcal{E}_{\text{o}}$). This *replaces* that ordinary STN subgraph with a minimal equivalent dispatchable STN. Next, any marked ordinary edges that were not removed by the STN dispatchability algorithm are removed at Line 26.

Lines 28-34 remove negative (ordinary or wait) edges emanating from any contingent TP, and non-negative ordinary edges terminating at any contingent TP. That's because in a dispatchable network, the presence of such edges necessarily imposes constraints on the corresponding activation TP that make the above-mentioned edges redundant.

Lines 35-42 remove wait edges that are dominated by alternative paths that are either ordinary or involving a different wait edge associated with the same contingent link.

**Complexity.** The worst-case time complexity of the $minDisp_{\text{ESTNU}}$ algorithm is dominated by the $k{+}1$ calls to Johnson's algorithm which has complexity $O(mn \log n)$, where $m \leq n^2$ is the maximum number of edges in the created STN. Therefore, the overall complexity is $O(kn^3)$.

## 4   Empirical Evaluation

We evaluated the performance of the $minDisp_{\text{ESTNU}}$ algorithm, comparing it against *Morris14* and $FD_{\text{STNU}}$. Our aim was to demonstrate: (1) the increase in computational cost required to minimize a dispatchable ESTNU, and (2) the reduction in the number of edges in the minimized network.

*Morris14* is the DC-checking algorithm proposed by Morris (2014), modified according to his high-level description to enable it to generate equivalent dispatchable networks for DC instances. We implemented the modified version for this evaluation. $FD_{\text{STNU}}$ is the fast dispatchable STNU algorithm presented by Hunsberger and Posenato (2022).

We implemented all algorithms in Java and ran them on a JVM 21 with 8GB of heap memory on a Linux computer with two AMD Opteron™ 4334@3.1 GHz (6200 BogoMIPS). Our implementations are available at (omitted for blind review).

For our evaluation, we used one of the benchmarks published by Posenato (2020). For each $n \in$

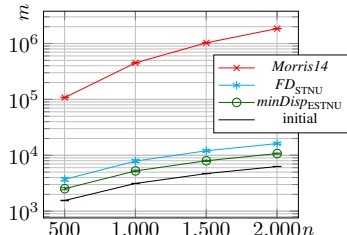

Figure 4: Number of edges, $m$, vs number of nodes, $n$

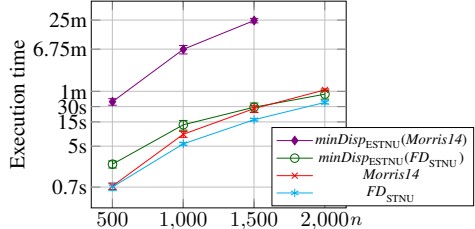

Figure 5: Execution time vs number of nodes, $n$

$\{500, 1000, 1500, 2000\}$, the benchmark contains 30 randomly generated DC STNUs, each having $n$ nodes, $n/10$ contingent links, and about 6 incident edges for each node (for a total of $m \approx 3n$ edges).

Each DC STNU was separately fed to the *Morris14* and $FD_{\text{STNU}}$ algorithms to generate a pair of equivalent dispatchable ESTNUs. Both ESTNUs were then fed to the *minDisp*$_{\text{ESTNU}}$ algorithm to check that the two minimized networks were the same. For the large majority of instances, they were, but for a few instances, the two minimized ESTNUs had some different edges. We discovered that simultaneity constraints among pairs of timepoints can result in trivially different, but equivalent minimized ESTNUs.

The plots in Figure 4 display the average numbers of edges in: the original STNU (black); the ESTNU generated by *Morris14* (red) and $FD_{\text{STNU}}$ (blue); and the minimized ESTNU generated by *minDisp*$_{\text{ESTNU}}$ (green). The error bars show 95% confidence intervals (difficult to see because the standard deviations are small). The average number of edges in the minimized networks is an *order of magnitude smaller* than in the ESTNUs generated by *Morris14* and *about 34% smaller* than in the ESTNUs generated by $FD_{\text{STNU}}$, confirming the importance of *minDisp*$_{\text{ESTNU}}$ for providing dispatchable networks that can be more efficiently executed.

Fig. 5 demonstrates the computational cost of generating minimal dispatchable ESTNUs. The results highlight the influence of the number of edges on the computing time, a factor that is absorbed by the $O(kn^3)$ theoretical worst-case complexity. Indeed, the *minDisp*$_{\text{ESTNU}}$ average execution time when *Morris14* gives the input instances (violet line) is an order of magnitude higher than the average execution time of *minDisp*$_{\text{ESTNU}}$ when $FD_{\text{STNU}}$ gives the input instances (green line). For $n = 2000$, the average execution time of *minDisp*$_{\text{ESTNU}}$(*Morris14*) is not reported because it was above the 30m timeout. Such a difference is equal to the difference in the average number of edges between the instances

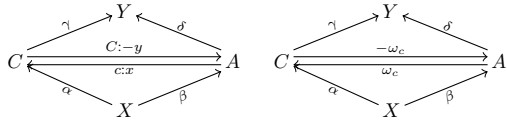

Figure 6: The simpler case covered by Lemma 2

determined by *Morris14* and $FD_{\text{STNU}}$ (see Figure 4) and is explained by the most expensive operation in *minDisp*$_{\text{ESTNU}}$ (i.e., the determination of all-pairs-shortest-paths) having complexity $O(mn \log n)$, where $m$ is the number of edges.

Another interesting result: for most instances with 2000 nodes, the execution time of *minDisp*$_{\text{ESTNU}}$($FD_{\text{STNU}}$) was smaller than *Morris14* without running *minDisp*$_{\text{ESTNU}}$.

## 5  Formal Analysis

This section presents a proof of correctness for the *minDisp*$_{\text{ESTNU}}$ algorithm. Throughout this section, we assume $\mathcal{G}$ is a dispatchable and weak-ord complete ESTNU.

**Definition 10.** We say that an edge $e$ in a projection $\mathcal{G}_\omega$ *derives from* a contingent link $(A, x, y, C)$ if $e$ is the projection of an LC, UC or wait edge associated with that link.

**Definition 11** (Needed). A contingent link $(A, x, y, C)$ is *needed* for $\text{maxd}(X, Y)$ in $\mathcal{G}$ if removing *all* of the labeled edges associated with that contingent link from $\mathcal{G}$ would change the value of $\text{maxd}(X, Y)$.

If no contingent link is needed for $\text{maxd}(X, Y)$, then its value is determined solely by ordinary edges in $\mathcal{G}$. Otherwise, there must be some contingent link $(A, x, y, C)$ that is needed to determine $\text{maxd}(X, Y)$.

**Lemma 1.** *If a contingent link $(A, x, y, C)$ is needed for $\text{maxd}(X, Y)$, then at least two labeled edges associated with $(A, x, y, C)$ are needed for $\text{maxd}(X, Y)$.*

*Proof.* Suppose only one labeled edge is needed.
  *Case 1: The LC edge $e = (A, c{:}x, C)$ is the only labeled edge associated with $(A, x, y, C)$ that is needed for $\text{maxd}(X, Y)$.* Let $\omega$ be any situation in which $e$ is needed. Then for any situation $\omega'$ that is the same as $\omega$ except possibly for the value of $C - A$, $e$ must also be needed, since otherwise there would have to be a path $P$ in $\mathcal{G}$ that does not include $e$ and such that $|P|_{\omega'} \leq \text{maxd}(X, Y)$. But then $|P|_\omega = |P|_{\omega'} = \text{maxd}(X, Y)$, contradicting that $e$ is needed in $\mathcal{G}_\omega$. But since $e$ is needed in $\mathcal{G}_{\omega''}$, where $\omega_c'' = y = |e_o|$, then the weak-ord edge $e_o$ contradicts the need for $e$.
  *Case 2: The only needed labeled edge $E$ is either the original UC edge $(C, C{:}{-}y, A)$ or some wait edge $(V, C{:}{-}v, A)$.* This case is similar to Case 1 except that $E$ must be needed in $\mathcal{G}_{\omega\dagger}$, where $\omega_c^\dagger = -x = |E_o|$, implying that the weak-ord edge $E_o$ contradicts the need for $E$. $\square$

**Lemma 2.** *If $(A, x, y, C)$ is needed for $\text{maxd}(X, Y)$, then the original LC and UC edges for $(A, x, y, C)$ are, by themselves, insufficient for determining $\text{maxd}(X, Y)$.*

*Proof.* If no wait edges are needed, then we have the simpler structure in Figure 6 where $|XACY|_\omega = \beta + \omega + \gamma$

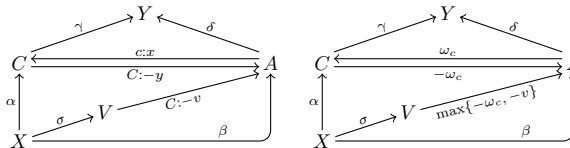

Figure 7: An ESTNU structure and its projection

$$f_1(\omega_c) = |XVAY|_\omega \quad = \sigma + \max\{-v, -\omega_c\} + \delta$$
$$f_2(\omega_c) = |XACY|_\omega \quad = \beta + \omega_c + \gamma$$
$$f_3(\omega_c) = |XVACY|_\omega = \sigma + \max\{-v, -\omega_c\} + \omega_c + \gamma$$
$$f_4(\omega_c) = |XCAY|_\omega \quad = \alpha - \omega_c + \delta$$

Table 2: The lengths of the paths in Figure 7

(increasing with $\omega$) and $|XCAY|_\omega = \alpha - \omega + \delta$ (decreasing with $\omega$). For both the LC and UC edges to be needed, the minimum path lengths across the entire range of $\omega \in [x, y]$ must use $|XACY|_\omega$ for smaller values of $\omega$, and $|XCAY|_\omega$ for larger values of $\omega$. This minimum takes on its maximum value at the intersection of the two plots (i.e., where $\beta + \omega + \gamma = \alpha - \omega + \delta$). This occurs when $\omega = (\alpha + \delta - \beta - \gamma)/2$. The value of both plots at that point is $((\alpha + \gamma) + (\beta + \delta))/2 = (|XCY|_\omega + |XAY|_\omega)/2 > \mathrm{maxd}(X, Y)$, which is a contradiction. Hence, the only way the contingent link $(A, x, y, C)$ can matter for the value of $\mathrm{maxd}(X, Y)$ is if at least one wait edge is needed. □

**Theorem 2.** *Suppose $(A, x, y, C)$ is a contingent link needed for $\mathrm{maxd}(X, Y)$ such that no other needed contingent link is constrained to have its activation TP occur after $A$. Then*

$$\mathrm{maxd}(X, Y) = \min\{\mathrm{d_o}(X, V) + \mathrm{maxd}(V, Y)$$
$$\text{such that } \exists (V, C{:} - v, A) \in \mathcal{E}_{\mathrm{ucg}}\}$$

*Proof.* Let $\omega'$ be any situation in which some paths, $P_1, P_2, \ldots$, each involving one or more edges derived from $(A, x, y, C)$, are needed to determine $\mathrm{maxd}(X, Y)$; and let $\omega'_c$ denote the value of $C - A$ specified by $\omega'$. By construction, the lengths of the $P_i$ paths in $\mathcal{G}_{\omega'}$ must be *strictly less* than the lengths of any other paths from $X$ to $Y$ in $\mathcal{G}_{\omega'}$.

Next, let $\Omega'_c$ denote the *family* of situations, where each situation in $\Omega'_c$ specifies the same duration as $\omega'$ for every contingent link *except* possibly $(A, x, y, C)$. If we let $\mu \in [x, y]$ be any possible value for $C - A$, then the family of situations $\Omega'_c$ is effectively parameterized by $\mu$; so we can let $\Omega'_c(\mu)$ denote the situation in $\Omega'_c$ for which $C - A = \mu$.

Since the lengths of the $P_i$ paths are less than the lengths of any other paths from $X$ to $Y$ in $\mathcal{G}_{\omega'}$; and their lengths vary piecewise-linearly with $\mu$ or $-\mu$, there must be some non-trivial open interval $(a, b)$ where $\omega'_c \in (a, b) \subseteq [x, y]$ and the lengths of the $P_i$ paths are all less than the lengths of any other paths from $X$ to $Y$ in any situation $\omega = \Omega'_c(\mu)$ for which $\mu \in (a, b)$. Without loss of generality, choose the smallest value of $a$ and the largest value of $b$ for which this property holds. Furthermore, since all relevant functions are continuous, we may include the endpoints of the interval $(a, b)$ with the understanding that the lengths of the $P_i$ paths may be less than *or equal* to the lengths of any other paths from $X$ to $Y$ in the situations $\Omega'_c(a)$ and $\Omega'_c(b)$. With this in mind, let $\Omega^*_c$ refer to the subset $\{\Omega'_c(\mu) \mid \mu \in [a, b]\}$.

Note: For convenience, in what follows, we use $\omega$ to refer to any situation in the restricted family $\Omega^*_c$; and we refer to the duration of $C - A$ by $\omega_c$ rather than $\mu$ (i.e., we effectively stipulate that $\Omega^*_c$ is parameterized by $\omega_c$, rather than $\mu$).

Let $\omega$ be any projection in $\Omega^*_c$. As illustrated in Figure 7, each simple shortest vee-path from $X$ to $Y$ in $\mathcal{G}_\omega$, which by construction must include edges derived from $(A, x, y, C)$, must have one of the following forms:[2] $XVAY, XACY, XVACY$ or $XCAY$, where the subpaths $XC, XV, XA, CY$ and $AY$ contain only ordinary edges or edges derived from *other* contingent links. Therefore, the lengths of those subpaths are *constant* across all situations in $\Omega^*_c$. In contrast, the lengths of the paths, $XVAY, XACY, XVACY$ and $XCAY$, are functions of $\omega_c$, as listed in Table 2 using names such as $f_1(\omega_c), f_2(\omega_c)$, etc.

Note that, by construction, $XCY$ and $XAY$ don't have any edges derived from $(A, x, y, C)$. Thus, using an argument similar to that in the proof of Lemma 1, they cannot be needed for $\mathrm{maxd}(X, Y)$ across the family $\Omega^*_c$. Therefore, $\mathrm{maxd}(X, Y) < |XCY|_\omega$ and $\mathrm{maxd}(X, Y) < |XAY|_\omega$.

From Table 2 it is apparent that the four length functions are each continuous, piecewise-linear and monotone (either non-decreasing or non-increasing). Therefore, if we define $f(\omega_c) = \min_{1 \le i \le 4}\{f_i(\omega_c)\}$ to be the *minimum* of the four length functions over $[a, b]$, then it too is continuous and piecewise-linear, although not necessarily monotone. Furthermore, the *endpoints* of each segment of the $f(\omega_c)$ function are necessarily at values of $\omega_c$ where at least two of the four length functions intersect. In what follows, we first compute the values of $f(\omega_c)$ at each possible intersection point, which completely determines the value of $\mathrm{maxd}(X, Y) = \max\{f(\omega_c) \mid \omega_c \in [a, b]\}$. We then show that for each such $\omega_c$, $f(\omega_c) = \min\{f_1(\omega_c), f_3(\omega_c)\} \le \mathrm{maxd}(X, Y)$. This will prove that it suffices, in general, to restrict attention to $f_1$ and $f_3$ (i.e., to the paths $XVAY$ and $XVACY$) to determine $\mathrm{maxd}(X, Y)$, which shall require only computing the single value of $\omega_c$ where those two paths intersect.

**Computing the intersection points.** To keep track of all possible points of intersection among the paths $XVAY, XACY, XVACY$ and $XCAY$, we use expressions such as $\tau_{ij}$ to denote the value of $\omega_c$ where $f_i(\omega_c) = f_j(\omega_c)$. For example, $\tau_{13}$ denotes the value of $\omega_c$ at which $f_1(\omega_c) = f_3(\omega_c)$ (i.e., where $|XVAY| = |XVACY|$). Intersection points that depend on the projected length of the wait edge (i.e., $\max\{-v, -\omega_c\}$) are given the superscript $v$ (for the case where $-v \ge -\omega_c$) or $w$ (for the case where $-\omega_c \ge -v$).

---

[2] Lemma 2 does not rule out that multiple wait edges might be needed across different situations. However, in the projection $\mathcal{G}_{\omega'}$, some wait edge $(V, C{:}{-}v, A)$ must minimize the length of the path from $X$ to $V$ to $A$ over some open interval, as described earlier. (If more than one wait yields the same minimum value, then disregard all but one.) Therefore, we may safely assume henceforth that only one wait edge is needed over the restricted family of situations $\Omega^*_c$.

It is easy, but tedious to compute the seven distinct points of intersection. For example, $f_1(\omega_c) = f_2(\omega_c)$ if and only if $\sigma + \max\{-v, -\omega_c\} + \delta = \beta + \omega_c + \gamma$. For $-v \geq -\omega_c$, we get $\omega_c = \sigma + \delta - v - \beta - \gamma$, which we notate as $\tau_{12}^v$.

Here's the full list: $\tau_{12}^v = \sigma + \delta - v - \beta - \gamma$, $\tau_{13} = \delta - \gamma$, $\tau_{12}^w = (\sigma + \delta - \beta - \gamma)/2$, $\tau_{14}^v = \alpha - \sigma + v$, $\tau_{34}^v = (\alpha + \delta - \sigma + v - \gamma)/2$, $\tau_{23}^w = \sigma - \beta$, and $\tau_{34}^w = \alpha + \delta - \sigma - \gamma$. Lemma 3, below, ensures that at each intersection point $\tau$, $f(\tau) = \min\{f_1(\tau), f_3(\tau)\} = \min\{|XVAY|, |XVACY|\} \leq \mathrm{maxd}(X, Y)$. Hence, all paths needed to determine $\mathrm{maxd}(X, Y)$ must include a wait edge. Also, if it were possible for two different waits, $(V_1, C:-v_1, A)$ and $(V_2, C:-v_2, A)$, to be needed in the same situation to determine $\mathrm{maxd}(X, Y)$, Lemma 3 ensures that they each achieve their maximum value at the same $\omega_c = \tau_{13} = \delta - \gamma$. Hence, both could be needed only if their maximums were the same. Therefore, using one needed wait $(V, C:-v, A)$ suffices; and all SSVPs from $X$ to $Y$ start with a prefix $XV$ that necessarily comprises only negative edges (since the wait edge is negative and $XV$ precedes it in a vee-path). But then any timepoint in $XV$ is constrained to occur after $A$ which, by the choice of $(A, x, y, C)$ implies that $XV$ cannot contain the activation TP for any *other* contingent link needed for $\mathrm{maxd}(X, Y)$. So, the value of $\mathrm{maxd}(X, V)$ cannot depend on any needed contingent links. Thus, $\mathrm{maxd}(X, V)$ only depends on ordinary edges, and $\mathrm{maxd}(X, Y) = \min_{(V, C:-v, A) \in \mathcal{E}_{\mathrm{ucg}}}\{\mathrm{d}_\mathrm{o}(X, V) + \mathrm{maxd}(V, Y)\}$. Hence, computing $\mathrm{maxd}(X, Y)$ reduces to computing $\mathrm{maxd}(V, Y)$ values for wait edges $(V, C:-v, A)$. □

**Lemma 3.** *For each intersection point $\tau$ listed earlier,*

$$f(\tau) = \min\{f_1(\tau), f_3(\tau)\} \leq \mathrm{maxd}(X, Y)$$

*Proof.* Due to space limitations, we only illustrate the proof for the case of $\tau_{12}^v$ (i.e., where $f_1 = f_2$ and $-v \geq -\omega_c$). If the minimum is $f_1(\tau_{12}^v) = f_2(\tau_{12}^v)$ or $f_3(\tau_{12}^v)$, then the result holds. Therefore, it suffices to show that the min value is *not* $f_4(\tau_{12}^v) = \alpha - \tau_{12}^v + \delta = (\alpha + \gamma) + (\beta - \sigma + v)$. If $\beta > (\sigma - v)$, then $f_4(\tau_{12}^v) > (\alpha + \gamma) > \mathrm{maxd}(X, Y)$. But if $\beta \leq (\sigma - v)$, then the ordinary path $XA$ dominates $XVA$ for all $\omega_c \in [a, b]$, contradicting the need for the wait edge. □

### Dealing with Nested Structures

As seen previously in Figure 3, the structures addressed by our algorithm can be *nested*. But Theorem 2 ensures that nesting can only occur in the paths, $CY$ or $AY$ (or both). Therefore, we can use an inductive argument to prove that the algorithm correctly computes every $\mathrm{maxd}(V, Y)$ value by analyzing paths starting with a wait edge $(V, C:-v, A)$; and therefore it correctly computes every $\mathrm{maxd}(X, Y)$ value.

**Theorem 3.** *Given any dispatchable ESTNU, Algorithm 1 correctly computes every $\mathrm{maxd}(X, Y)$ value.*

*Proof.* We begin with the following observations. First, in any SSVP, there can be no repeat nodes or edges; therefore, the edges that derive from our chosen $(A, x, y, C)$ cannot appear in either $CY$ or $AY$. Therefore, at most $k$ contingent links can be needed to determine any given $\mathrm{maxd}(X, Y)$

value. Since the order of nesting is typically not known in advance, the algorithm performs $k$ outer iterations to ensure that every possible nesting order can be accommodated.

Second, for any contingent links needed for $\mathrm{maxd}(C, Y)$ or $\mathrm{maxd}(A, Y)$, there must be durations that ensure that $|CY|_\omega = \mathrm{maxd}(C, Y)$ and $|AY|_\omega = \mathrm{maxd}(A, Y)$. This holds even if $CY$ and $AY$ share a common suffix. And these durations may be chosen independently of the durations of any other contingent links. And since these maximum values for $|CY|$ and $|AY|$ can only result in increased values for $|VACY|$ and $|VAY|$, replacing $CY$ and $AY$ by the ordinary edges $(C, \mathrm{maxd}(C, Y), Y)$ and $(A, \mathrm{maxd}(A, Y), Y)$ cannot change the value of $\mathrm{maxd}(V, Y)$.

Since the values of $\mathrm{maxd}(C, Y)$ and $\mathrm{maxd}(A, Y)$ cannot need as many contingent links as the value of $\mathrm{maxd}(V, Y)$, because they cannot use $(A, x, y, C)$, the inductive hypothesis ensures that the algorithm correctly computes $\mathrm{maxd}(C, Y)$ and $\mathrm{maxd}(A, Y)$ and, therefore, it correctly computes $\mathrm{maxd}(V, Y)$. □

**Lemma 4.** *For a dispatchable ESTNU $\mathcal{G}$, and contingent timepoint $C$, removing negative (ordinary or wait) edges emanating from $C$, and non-negative ordinary edges pointing at $C$ cannot threaten the dispatchability of $\mathcal{G}$.*

*Proof.* Omitted due to space limitations. □

**Theorem 4.** *Algorithm 1 computes an equivalent dispatchable ESTNU having a minimal number of edges.*

*Proof.* Suppose the output $\mathcal{G}$ of Algorithm 1 contains an ordinary edge $e = (X, \delta, Y)$ that is not needed for dispatchability. Then in every projection $\mathcal{G}_\omega$, there must be some shortest vee-path $\mathcal{P}$ from $X$ to $Y$ that does not use $e$ such that $|\mathcal{P}|_\omega \leq \delta$. But then $\mathrm{maxd}(X, Y) \leq \delta$ and the algorithm marks $e$ for removal, contradicting that $e$ in the output $\mathcal{G}$.

Next, suppose $E = (V, C:-v, A)$ is a wait edge in the output $\mathcal{G}$ that is not needed for dispatchability but not marked for removal. This happens when all of the following hold: (1) $\mathrm{maxd}(V, A) > -v$; (2) $\mathrm{maxd}(V, C) > 0$; and (3) for each wait $(U, C:-u, A) \in \mathcal{E}_{\mathrm{ucg}}$, $\mathrm{maxd}(V, U) > 0$ or $\mathrm{maxd}(V, U) - u > \max\{-y, -v\}$. We will show that there must be a value $\omega_c < v$ such that in that projection, every alternative path from $V$ to $A$ has length greater than $-\omega_c = |(V, C:-v, A)|$. First, note that for any $(U, C:-u, A)$, if $\mathrm{maxd}(V, U) > 0$, then $|VUA| > \max\{-u, -\omega_c\} > -\omega_c$. Next, let $-u^*$ be the minimum value of $\mathrm{maxd}(V, U) - u$ for all waits $(U, C:-u, A)$ where $\mathrm{maxd}(V, U) - u > \max\{-y, -v\}$. Then $-u^* > -v$. Let $-u^\dagger = \min\{-u^*, \mathrm{maxd}(V, A)\} > -v$. And then let $\omega_c$ be any point in $(u^\dagger, v)$. It is not hard to check that the length of every alternative path from $V$ to $A$ (i.e., $VA_o$, $VCA$ or any $VUA$) is greater than $-\omega_c = |E|$. Hence, $E$ is needed. □

## 6  Conclusions

This paper presented a new algorithm for generating equivalent dispatchable ESTNUs having a minimal number of edges. The empirical evaluation demonstrated its effectiveness in reducing the number of edges. The formal analysis proved the algorithm's correctness.

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
