# OpenReview forum: "Converting Simple Temporal Networks with Uncertainty into Minimal Equivalent Dispatchable Form"
_icaps-conference.org/ICAPS/2024/Conference — ICAPS 2024_

### Official Review · Reviewer_2mBv · 2024-01-17

**Significance And Importance:** 1
**Soundness:** 3
**Novelty:** 2
**Clarity:** 2
**Confidence:** 4

**Weaknesses:**

-1: Major weaknesses requiring significant work to be addressed for the paper to be accepted.

**Contributions Of The Paper:**

This paper presents a novel algorithm for converting any dispatchable (dynamic controllable) Simple Temporal Network with Uncertainty (STNU), for which a dynamic execution strategy can be generated incrementally, into an equivalent dispatchable network having a minimal number of edges. The property of being dispatchable seems to be essential for some practical applications.

Existing algorithms can generate equivalent dispatchable STNUs, but with no guarantee about the number of edges in the output STNU.

The paper thus introduces an algorithm such that when given a dispatchable (Extended) STNU as input, it outputs an equivalent dispatchable ESTNU with a minimal number of edges. The explanation of the algorithm is very technical. The related section is organized such that some special cases are presented at the beginning before presenting the actual algorithm. I am wondering whether the pseudo-code of the algorithm could come first to improve the readability of the section, which is limited.

Some technical questions on this part follow:

- What does it mean "equivalent minimized ESTNUs" ?

- Why is it important to be "dispatchable" ? It should be clarify earlier, also wrt what does it mean

An experimental analysis has been performed on randomly generated benchmarks to demonstrate: (1) the increase in computational cost required to minimize a dispatchable ESTNU, and (2) the reduction in the number of edges in the minimized network.

Of course, the analysis confirms these points. However, they are expected results. The authors should have, instead, in my opionion stressed, also with an experimental evaluation, what is the importance/usefulness of having a minimal network (for practical applications).

Section 5 contains a proof of the correctness of the introduced algorithm. I believe this section should come right after Section 3, and then experiments.

To sum up, the paper presents and implements an algorithm for minimizing the number of edges in dispatchable STNU. Experiments of random benchmarks show expected results, and some dimensions are missed. Organization and clarity of the paper should be improved.

========== POST-REBUTTAL =============

The authors replied in a quite satisfying way to my comments, even if all major points I raised, which are shared with other reviewers, and outlined in the meta-review, basically remain.

=====================================


Minor points


pg 1

"An STNU" -> "A STNU?" This may recur in the paper.

pg 2

"Others (citation omitted for blind review) " - this means that the authors of these papers are among those working on STNU and not cited in the paper? For kowing the authors, it should be enough now to search for a paper in which they "formally defining a real-time execu- tion algorithm for ESTNUs and proving that it is guaranteed to successfully execute any dispatchable ESTNU".

pg 3

"Then in every projection" -> "Then, in every projection"

"monotonic" -> "monotone" ?

pg 4

"is outside of the interval " -> "is outside  the interval "

pg 6

**Ethical Considerations:**

(1) Not Applicable: The paper does not have any ethical considerations to address

**Nomination For Best Paper:**

No

**Overall Evaluation:**

-1: (weak reject)

**Questions For Authors:**

See review in the contribution of the paper section.

**Reproducibility:**

0: N/A - nothing to reproduce.

**Strengths Of The Paper:**

The paper contributes to both algorithm and practice in the area of STNU.

**Weaknesses Of The Paper:**

- Limited contribution
- Experiments
- Organization and clarity

---

> ### Author Rebuttal · Authors · 2024-01-26
>
> RE: "Significance And Importance", we respectfully urge you to reconsider your review, especially in view of the review by bLAQ,
> who scored it "3: Substantial contribution or strong impact" and nominated our paper for "Best Paper".
>
> RE: "Why is it important to be dispatchable?" The literature on dispatchability over the past 25 years is unequivocal: during execution, it is important to limit computation time. "EXEC is interested in doing the minimal amount of computation required in order to keep consistency of the plan during execution ... Because propagation time has a major impact on the EXEC latency, it is essential that propagation time be minimized" (Issues in Temporal Reasoning for Autonomous Control Systems, Muscettola, Morris, Pell and Smith, 1998). Dispatchability is crucial for time-critical missions (originally motivated by NASA space missions). We will make this clearer in the paper.
>
> We use "equivalent minimized ESTNUs" to mean two possibly different ESTNUs that are equivalent, dispatchable, and both having a minimal number of edges.
>
> RE: "Of course, the analysis confirms these points. However, they are expected results." The purpose of the empirical
> evaluation is to demonstrate the average performance of our new algorithm across a published benchmark, as opposed
> to the theoretical worst case of $O(kn^3)$.   We focused on the performance
> of our new algorithm, since that was the contribution of this paper.
>
> The omitted citation is for a paper that has been *accepted* for publication, but has not yet appeared. Therefore, a Google search does not find it. We believe that omitting this citation is proper according to ICAPS procedures for blind review.
>
> RE: Experimental evaluation showing importance of minimal network for practical applications. We can provide an empirical evaluation *and* a theoretical analysis of the performance of the real-time execution (RTE) algorithm on the original ESTNU versus the minimal dispatchable ESTNU. The RTE alg uses a min priority queue GLB to track min lower bound among timepoints (TPs) that can execute "next". When a contingent TP executes, RTE deletes associated waits which may reduce lower bounds for some TPs, requiring decreaseKey ops for GLB. Total cost for GLB: $O(w+n\log n)$, where w is num wait edges. (Lower bounds from ordinary propagation are computed before TP enters GLB queue.) Total cost for Upper-Bound queue (no waits) is $O(m + n\log n)$ (m is num ordinary edges).  See Rebuttal 1 for related info.

---

### Official Review · Reviewer_bLAQ · 2024-01-17

**Significance And Importance:** 3
**Soundness:** 3
**Novelty:** 3
**Clarity:** 3
**Overall Evaluation:** 2
**Confidence:** 3

**Weaknesses:**

1: Minor weaknesses that are easily fixable.

**Contributions Of The Paper:**

This paper discusses how to transform an STNU into a minimal dispatchable network.  STNUs are simple temporal networks augmented with uncertain links.  A dispatchable network is a network in which the network can be controlled by simply updating the neighbors (in the network) of the new events.  Finally, a dispatchable network is minimal if it does not include unnecessary edges, thereby guaranteeing minimal work when a new event is observed.

**Ethical Considerations:**

(1) Not Applicable: The paper does not have any ethical considerations to address

**Nomination For Best Paper:**

Yes

**Questions For Authors:**

- Definition 7 (Dispatchable ESTNU) defines dispatchability of a ESTNU as a property that is difficult to assess (it requires, at first glance, to verify the property of an infinite number of projections).  Is there a simpler (to verify) property similar to Theorem 1?

- Is my understanding correct that the structure of Figure 1 appears everywhere in the network?

**Reproducibility:**

4: Authors promise to release code and domains (whichever apply).

**Strengths Of The Paper:**

- The paper addresses an important problem that has remained open for a decade now.  Minimal dispatchability of STNs was solved in 1998.  Dispatchability of STNUs was solved in 2014 if I am not mistaken.

- The paper is very well written.  It provides good examples (although, as will be obvious in the next section, I haven't understood everything) and is very didactic.

- I appreciate that the experimental section is provided before the proofs as it allows us to evaluate the impact / benefits of the approach before doing the hard work of understanding why it works.

**Weaknesses Of The Paper:**

There are several issues in the paper most of which can be easily fixed.

- Definition of dispatchability.

Dispatchability is first defined as the property of an STN, which states that a specific
algorithm always succeeds when applied to the STN, and then Th1 shows
that dispatchability is equivalent to the existence of shortest
vee-paths in the graph.  I think the authors should explicitly state
that they will use this characterisation of dispatchability as its
formal definition (it was unclear to me as I read the paper, and I was
constantly expecting a formal definition of dispatchability).
(This is especially true given that the definition of dispatchability of
an ESTNU (Def. 7) is based on properties of the graph and not on the
property of an algorithm.)

The solution to this issue might be as simple as first explaining the TD algorithm (lines 105 and next), and only after explaining that dispatchability is the property that the TD algorithm always succeeds.

- Did I understand correctly that the example of Fig. 1 is a structure that appears all the time in an STNU?  I assumed that it was an example used to introduce some notions; I now believe that it is essentially the only structure that will be encountered in the network (together with that of Figure 3) and that the only aspects of the figure that will differ in a real instance are the specific numbers that will be attached to each edge.

If I am correct, then I would like this to be made explicit.

Misc.

Line 26
"representating" -> representing

Line 52, I do not understand the sentence ``such a strategy requires
exponential space''.  Exponential in what?  Also, what would the
execution strategy look like?  I expect for instance that the execution
strategy might specify things like: ``x units of time after observing o,
execute action a'', but the `language' used to represent the execution
strategy might become more elaborate: ``You may execute action a in the
time interval [max(something,something),min(something,something)]''.
Do you have a good reference that explains what you have in mind?
What do you mean by "*exceedingly* impractical"?

The (citation omitted/blind review) references are very unfortunate.  I
understand (and appreciate) that the authors are worried that their
submission might not be accepted.  However, how am I supposed to do my
job as a reviewer if the references are not available?  If the work is
currently under review (which I assume is the case, because already
published stuff wouldn't be a problem here), it seems that the work is
done incrementally (publish a paper on the strategy, then publish a
paper on improving the representation).  Why not publish everything in a
single paper?

Line 65: ``presented an algorithm for creating an equivalent
dispatchable STNU, called FDSTNU .''  This sentence is weird.
``Equivalent'' to what?  Do you mean ``an algorithm that takes an STNU
and creates an equivalent dispatchable STNU?''

Definition 6, technically, $\omega_i$ is not defined.  What has been defined was $\omega_c$.  Maybe the notation before the definition should be amended.

Line 228, there is an implicit assumption that there cannot be
subset-minimal dispatchable ESTNU that are not cardinality-minimal ones
(for instance, there might be an input ESTNU with three edges {a,b,c}
such that the sub-ESTNUs {a,b} and {c} are dispatchable, and  none of
their proper sub-ESTNUs are).  I am willing to believe that this is
true, but it would be nice to see a discussion about that.
* From memory of the work on STNs, there can be multiple
cardinality-minimal dispatchable STNs.  For instance, one might have the
choice between keeping edge (a->c) and (a->b) if the label on these
edges is the same.

After Def. 8, it is worth explaining that maxd(V,Y) will be used to
prune the edge between V and Y.  I believe that this is the purpose of
the paragraph Line 242-247, but maxd(V,Y) is not mentionned in this
paragraph.

I believe it's worth explaining why the structures are nested.  Do I
understand correctly that the nested comes from the fact that the edge
A2 to A has a negative upper-case edge, and that there cannot be a
negative upper-case edge from A to A2 because there would then be a
negative cycle, hence inconsistency?

Definition 9, unless I missed it, I am not sure that $\le$ has been defined for edges.  I presume this means two edges over the same vertices but the distance of $e$ is smaller than that of $e'$.

Line 415, why isn't the complexity O(kn3 log n)?

---

> ### Author Rebuttal · Authors · 2024-01-26
>
> First of all, thanks very much for recognizing the importance of our paper's contribution: "non-trivial advances over the current state-of-the-art"; and nominating it for Best Paper. As you noted, dispatchability for STNs was solved in 1998 (including finding dispatchable STN with min num edges); and dispatchability for ESTNUs was solved in 2014; but since then, finding dispatchable ESTNU with min num edges has been an open problem--until now.
>
> We will re-work the presentation of the proofs to make them clearer.
>
> RE: defining dispatchability, we agree that it's best to first define STN dispatchability as a performance guarantee RE: the TD algorithm, and then give Morris' equivalent characterization in terms of shortest vee-paths (our Theorem 1). For ESTNUs, Morris did the opposite: he first defined dispatchability as a graphical property (our Defn 7) then asserted (without proof) that a dispatchable ESTNU gives a performance guarantee for a roughly-sketched execution alg.
>
> The omitted citation is for a paper that has been accepted for publication but is not yet out.  It formally defines a real-time execution (RTE) alg for ESTNUs and proves Morris' conjecture. It does not address finding disp ESTNU with min num edges. This paper does.
>
> The goal of ESTNU dispatchability is the performance guarantee, but our proofs use the equivalent defn (Defn. 7). We will clarify this.
>
> The graph in Fig. 1 is presented as the canonical case where distinct shortest
> vee-paths from different STN projections combine to entail a stronger ordinary constraint that
> holds across all projections. It's not that this structure arises a lot. It's just that
> this is the only way that this can happen; so it is the only kind of special structure
> that our algorithm must explore.
>
> RE: Nested structures. The diamond $A_2$-A-C-Y creates a stronger ordinary edge $A_2Y$ that is then
> used by the outer diamond V-$A_2$-$C_2$-Y. This kind of nesting is rare, but the algorithm must
> check for it.
>
> For any STNU, a dynamic strategy, if fully written out, may have exponentially many conditional
> branches (exponential in the number of contingent links, $k$) because different observed durations
> may lead to different execution decisions. That is why it is only practical to incrementally
> generate the strategy in response to observations as they occur.
>
> RE: FD-STNU. That algorithm takes an STNU as input and, if it is DC, generates as output an
> equivalent dispatchable ESTNU.
>
> RE: $O(k n^3)$.  See Rebuttal 1.

---

### Official Review · Reviewer_knGA · 2024-01-22

**Significance And Importance:** 2
**Soundness:** 2
**Novelty:** 2
**Clarity:** 2
**Overall Evaluation:** 2
**Confidence:** 3

**Weaknesses:**

0: Minor weaknesses requiring some work to be addressed for the paper to be accepted.

**Contributions Of The Paper:**

STNUs are called dynamically controllable if there exists an executing strategy for all possible situations. This paper elaborates on two state-of-the-art algorithms (Morris14, FD_STNU) that take as input an STNU and return an equivalent dispatchable (extended) ESTNU. The new algorithm (minDisp_ESTNU) takes as input an ESTNU and returns an equivalent network with a minimal number of edges. The authors claim that the (reduction in the) number of edges in such an ESTNU directly impacts the computations during execution.
The paper starts with a background on all relevant concepts, after which the novel algorithm is introduced in a dedicated section, accompanied by some illustrative examples to explain the key concepts of the algorithm.

The algorithm first inserts weak-order edges to the ESTNU. Then, all possible simpler structures are explored, to see whether some edges can be removed. This concept of looking for simple structures is explained in Figure 2. These special structures are made explicit by inserting ordinary edges. The second part of the algorithm is dedicated to determining which edges to remove. The complexity is O(kn^3), where k is the number of actions with uncertain durations and n is the number of time points.

The minDisp_ESTNU algorithm is tested empirically in combination with both Morris14 and FD_STNU and compared against the two algorithms without the minDisp_ESTNU post-processing. The empirical evaluation analyzes the increase in computational costs when the edge-minimizing algorithm is applied, and the reduction in the number of edges of the minimized network.

The paper finalizes with a formal analysis to ensure the correctness of the algorithm with concrete proofs. Multiple theorems and lemmas are presented together with their proofs.

**Ethical Considerations:**

(1) Not Applicable: The paper does not have any ethical considerations to address

**Nomination For Best Paper:**

No

**Questions For Authors:**

1. Could you provide the theoretical justification for the claim that reducing the number of edges reduces the computation time during execution?
2. By how much does the number of edges reduce the computation time during execution in your experiments?
3. Could you explain how you came to O(kn^3)? So, could you explain why the k+1 calls of Johnson dominate the complexity, and not the subsequent nested for loops?
4. Could you provide the proof of Lemma 4 in supplementary material?
5. Could you provide the needed connections between Section 3 and Section 5 to improve the clarity of the paper? So instead of, “will be shown” in Section 3, please refer to exact Lemmas/Theorems on which these claims are based.

POST-REBUTTAL:
Thank you for your answers. We've updated our scores, assuming you'll indeed follow-up as described in your rebuttal, including the promised plot.

**Reproducibility:**

4: Authors promise to release code and domains (whichever apply).

**Strengths Of The Paper:**

Relevance/originality: The topic of the paper is a relevant fit for ICAPS, because it provides an incremental improvement on state-of-the-art methods in the context of temporal planning. The authors introduce a novel idea to minimize the number of edges of a DC ESTNU.

Clarity/scholarship: The introduction nicely captures relevant work and summarizes the core idea of the authors’ contribution. The background sections introduce standard notation and definitions that help the reader understand important concepts related to STNUs. Even though many definitions and notations are introduced in these first sections, the authors manage to provide clarity, mainly because of their readable notation and consistent terminology.  The relation with related literature is clear for the reader.

Presentation: The authors build up pieces of their algorithm by using explanatory illustrations (e.g. Figures 1 and 3) that are accompanied by a textual explanation at the beginning of Section 3. This helps the reader to understand the core ideas used in their method and that is much appreciated. The goals of the experiments are well-explained, and the results are easy to interpret.

**Weaknesses Of The Paper:**

Significance: It is stated in the abstract and introduction that minimizing the number of edges reduces computation time during execution. This is the main motivation provided by the authors to claim the significance of their contribution. However, for the reader, it is unclear what the relation is between the number of edges and the computation time during execution. The statement is neither theoretically justified nor empirically evaluated.  Therefore, it cannot be judged whether the obtained results (reduction in number of edges) significantly improve the computation time for execution. This is an important concern that needs to be addressed for acceptance.

Clarity: In Section 3, multiple comments are included such as “as will be seen” (line 299), and “as will be shown” (e.g. line 304). I assume that the authors mean that they will further in the paper (Section 5 I believe) will formally explain why these statements are true. It might be helpful to already refer to the corresponding theorems/lemmas, to help the reader navigate through the different components of the algorithm and corresponding proofs. I find it rather hard to now navigate through Section 5 because I miss the connection with Section 3.
Also within Section 3 the subsection “Nesting of Special Structures” misses a bit of introduction and connection to the rest of the paper.
I think there is potential to improve the clarity of the paper by connecting the illustrative examples, subparts of the algorithm, and the corresponding theorems and proofs in a proper way.

Technical soundness: The authors claim that the complexity of the algorithm runs in O(kn^3) worst-case time, where k is the number of actions with uncertain durations, and n is the number of timepoints because it is dominated by the k+1 calls to Johnson's algorithm (line 9 /10 of pseudocode). For a non-expert reader, it is a bit hard to follow why this is correct, mainly why it is true that the complexity is dominated by the k+1 calls to Johnson’s algorithm and why it is true that m <= n^2.

---

> ### Author Rebuttal · Authors · 2024-01-26
>
> RE: significance, please see review by bLAQ who nominated our paper for "Best Paper".
>
> Thanks for suggesting to compare run-time of the real-time execution (RTE) alg on the original ESTNU vs. the equivalent dispatchable ESTNU with min num edges. We'll add a plot for this. We can also analyze theoretically, as follows. For m ordinary edges, RTE does at most m propagations of lower/upper bounds at total cost of O(m).  RTE also uses O(n) max priority queues to track max wait times for enabled timepoints, requiring O(w) Delete or ExtractMax ops, where w is total num of wait edges.  Each op costs O(k), where k is num contingent links.  So total cost is $O(w\log k)$. Our algorithm minimizes both m and w, and so reduces computation by RTE during execution.  (More info at end of Rebuttal 3.)
>
> We'll clarify connections between Sections 3 and 5 by referring to Lemmas and Theorems by number.
>
> RE: complexity of $\mathrm{minDisp}_\mathrm{ESTNU}$ alg., we will
> more clearly justify the $O(kn^3)$ result, as follows.
> First, the number of edges $m$ in any ESTNU satisfies $m \leq n^2$.
> So the complexity of $k+1$ calls to Johnson's alg. is $O(k m n + kn^2\log n) \leq O(kn^3)$, since $m \leq n^2$ and $\log(n) \leq n$.
>
> RE: the nested FOR loops:  there are $k$ iterations of FOR loop at line 9;
> $k$ iterations of the FOR loop at line 12; and $n^2$ iterations of the FOR loop at line 13
> ($n$ choices for $V$, $n$ choices for $Y$, and a constant-time check for a wait edge).
> So the nested FOR loops overall have complexity: $O(k^2n^2) \leq O(kn^3)$, since $k \leq n$.
>
> Proof of Lemma 4. Case 1: $(Y,d,C)$ is an ordinary edge
> where $d \geq 0$ and $C$ is contingent. The only role for a non-negative edge during execution
> is to propagate upper bound to its target timepoint. But the target here is contingent $C$ which
> the executor does not control, so any tightening of
> its upper bound would threaten the network's dynamic controllability,
> contradicting dispatchability. So this edge can play no meaningful role during
> execution.
>
> Case 2: $(C,d,X)$ is an ordinary edge where $d < 0$ and $C$ is contingent.
> A negative edge plays two roles during execution: determining whether its source
> timepoint is enabled and propagating a lower bound to its source. But the
> source here is the contingent $C$, to which the RTE alg. does not apply the enablement requirement.
> And, like in Case 1, any tightening of $C$'s lower bound would threaten the network's dynamic controllability,
> contradicting dispatchability.

---

### Meta-Review · Area_Chair_WdFY · 2024-02-06

**Recommendation:** Accept (Poster)
**Confidence:** 4

**Metareview:**

The paper elaborates presents an algorithm that transforms an (E)STNU into an equivalent one with a minimal number of edges. This is important for the execution of dynamic strategies, because the execution algorithm time is directly influenced by the size of the network.

There was considerable discussion on this paper.

Specifically, three points were raised:

1. Limited contribution: one of the reviewers thinks that finding the minimal dispatchable network is a very incremental contribution. The other reviewers argued that this is non-trivial and significant enough for a conference paper.
2. Experimental analysis: all experiments are on random networks, but the authors stress in the paper the importance for practical applications, and no experiments on real data are reported. In the rebuttal, the authors promised to provide a plot showing the benefits in term of execution time. We recommend to provide this data, but the majority of reviewers agreed that the paper is acceptable even without this promised contribution.
3. Organization and clarity: the organization of the paper with teh formal analysis at the end was criticized, but the reviewers do not see this as a fatal flaw.

In sumamry, we could not reach a consensus among the reviewers, but it seems to me that there are no knocking-down arguments for the paper, so I am recommending acceptance.

**Ethical Considerations:**

(1) Not Applicable: The paper does not have any ethical considerations to address